# Optimized residue analysis method for broflanilide and its metabolites in agricultural produce using the QuEChERS method and LC-MS/MS

Hyun Ho Noh[1], Chang Jo Kim[1], Hyeyoung Kwon[2], Danbi Kim[1]*, Byeong-chul Moon[1], Sujin Baek[1], Min-seok Oh[1], Kee Sung Kyung[3]

1 Chemical Safety Division, Department of Agro-Food Safety and Crop Protection, National Institute of Agricultural Sciences, Rural Development Administration, Wanju, Republic of Korea, 2 Planning and Coordination Bureau, Rural Development Administration, Jeonju, Republic of Korea, 3 Department of Environmental and Biological Chemistry, College of Agriculture, Life and Environment Science, Chungbuk National University, Cheongju, Chungbuk, Republic of Korea

* danbi6334@korea.kr

**Data Availability Statement:** All relevant data are within the paper.

## Abstract

Since broflanilide is a newly developed pesticide, analytical methods are required to determine the corresponding pesticide residues in diverse crops and foods. In this study, a pesticide residue analysis method was optimized for the detection and quantification of broflanilide and its two metabolites, DM-8007 and S(PFH-OH)-8007, in brown rice, soybean, apple, green pepper, mandarin, and kimchi cabbage. Residue samples were extracted from the produce using QuEChERS acetate and citrate buffering methods and were purified by dispersive solid-phase extraction (d-SPE) using six different adsorbent compositions with varying amounts of primary secondary amine (PSA), $C_{18}$, and graphitized carbon black. All the sample preparation methods gave low-to-medium matrix effects, as confirmed by liquid chromatography–tandem mass spectrometry using standard solutions and matrix-matched standards. In particular, the use of the citrate buffering method, in combination with purification by d-SPE using 25 mg of PSA and a mixture of other adsorbents, consistently gave low matrix effects that in the range from −18.3 to 18.8%. Pesticide recoveries within the valid recovery range 70–120% were obtained both with and without d-SPE purification using 25 mg of PSA and other adsorbents. Thus, the developed residue analysis method is viable for the determination of broflanilide and its metabolites in various crops.

## Introduction

In Korea, a positive list system, similar to systems in the United States, Europe, and Japan, was introduced in January 2019, setting a detection limit of 0.01 mg kg$^{-1}$ for any unregistered pesticide in agricultural products [1]. It is expected that the maximum residue limits will be frequently exceeded in agricultural products, in part because the pesticides used to treat the first

**Funding:** Initials of the author who received award: DK This study was funded by the Research Program for Agricultural Science & Technology Development, National Institute of Agricultural Sciences, Rural Development Administration, Korea (http://www.naas.go.kr; grant number: PJ013594012019) to NAAS, RDA. The funder had no role in study design, data collection and analysis, decision to publish, or preparation of the manuscript.

**Competing interests:** The authors have declared that no competing interests exist.

crop in a field can remain in the soil and be translocated to the second crop [2], and because aerial spraying can cause pesticides to drift unpredictably [3]. To minimize these problems, it is necessary to confirm that the residual pesticide levels in agricultural products do not exceed these limits. Furthermore, multiresidue analysis techniques for detecting multiple residual pesticides in agricultural products should be applied to newly developed pesticides. Various rapid and accurate methods have been established for such analyses.

A pesticide residue analysis method developed by the US Food and Drug Administration in the 1960s has been commonly applied to organochlorine pesticides [4]. However, this method, which involves liquid–liquid partitioning and adsorption chromatography in an open column, not only generates a large amount of waste but also requires a long analysis time and is thus expensive [5]. To minimize the quantity of organic solvent required, quicker and more effective analytical methods for residual pesticides have been developed. Among them, the QuEChERS (quick, easy, cheap, effective, rugged, and safe) method, developed by Anastassiades et al. [6], is currently the most widely used. In this method, residues are extracted from agricultural products using acetonitrile and $MgSO_4$, purified by dispersive solid-phase extraction (d-SPE) using adsorbents (e.g., primary secondary amine (PSA), $C_{18}$, and graphitized carbon black (GCB)), and then analyzed using gas chromatography–tandem mass spectrometry or liquid chromatography–tandem mass spectrometry (LC-MS/MS) [4]. $MgSO_4$ is used to separate water from the organic solvent and the various adsorbents are used to remove different kinds of interfering compounds. In particular, PSA removes polar organic acids, polar pigments, and some sugars and fatty compounds, GCB removes sterol and pigments like chlorophyll, and $C_{18}$ removes nonpolar compounds like lipids [4, 7].

The test pesticide broflanilide (*N*-[2-bromo-4-(perfluoropropan-2-yl)-6-(trifluoromethyl) phenyl]-2-fluoro-3-(*N*-methylbenzamido)benzamide) is a meta-diamide organic halide developed by Mitsui Chemicals Agro and Badische Anilin und Soda Fabrik (BASF) [8] and used to control insect pests (e.g., *Lepidoptera*) that eat the leaves of pulse crops, cereals, fruits, and vegetables, as well as to control ant, fly, and cockroach infestations [9]. The above pesticide, featuring one bromine atom and 11 fluorine atoms in its molecular structure and having a high log P (partition coefficient) [10], has two metabolites with similar structures, namely DM-8007 (3-benzamido-*N*-[2-bromo-4-(perfluoropropan-2-yl)-6-(trifluoromethyl)phenyl]-2-fluoro-benzamide) and S(PFH-OH)-8007 (*N*-[2-bromo-4-(1,1,1,3,3,3-hexafluoro-2-hydroxypropan-2-yl)-6-(trifluoromethyl)phenyl]-2-fluoro-3-(*N*-methylbenzamido)benzamide) [8]. Notably, DM-8007 has greater insecticidal activity than its parent compound [11].

A pesticide residue analysis method for broflanilide and its two metabolites in soil was reported by An et al. [8]. However, analysis methods for residual pesticides in crops and/or foods have not been widely reported. Moreover, since specific crops and/or foods may require different pesticide residue analysis methods, a variety of analytical methods should be developed and distributed. To this end, herein we established a rapid and efficient method for the analysis of broflanilide and its two metabolites in various test crops based on an optimized QuEChERS method.

## Materials and methods

### Test pesticide and test produce

The test pesticides were the insecticide broflanilide and its two metabolites, DM-8007 and S (PFH-OH)-8007. The test crops were rice (*Oryza sativa* L.), soybean (*Glycine max*), apple (*Malus pumila* Mill.), green pepper (*Capsicum annuum* L.), mandarin (*Citrus unshiu* Markovich), and kimchi cabbage (*Brassica rapa* L. ssp. *pekinensis*). These products were chosen because they are widely consumed in Korea and encompass different food groups [12]. In

Korea, brown rice is typically used, rather than white rice, for pesticide residue analysis. The untreated samples were bought at an environmentally friendly agricultural produce market, Chorocmaeul (www.choroc.com) in Wanju, Korea.

## Reagents and materials

Broflanilide (99.67% purity), DM-8007 (99.86% purity), and S(PFH-OH)-8007 (99.02% purity) were obtained from the Dongbang Agro Corporation (Korea). LC-gradient-grade acetonitrile for sample extraction and methanol were purchased from J. T. Baker (USA). HPLC-grade ammonium formate (>98%) and formic acid (>98%) used in the LC mobile phase were purchased from Merck (Germany) and Sigma-Aldrich (USA), respectively. The QuEChERS extraction pouch for sample extraction and the d-SPE tube for sample purification were purchased from Agilent Technologies (USA). A Combi-514R centrifuge (Hanil Scientific Inc., Korea) and a 1600 MiniG extractor (SPEX SamplePrep, USA) were used during sample preparation.

## Standard solutions

To prepare a 1,000 mg $L^{-1}$ stock solution, an appropriate amount of broflanilide, DM-8007, or S(PFH-OH)-8007 (20.07, 20.03, or 20.20 mg, respectively) was dissolved in 20 mL of acetonitrile. The stock solutions were combined to prepare a 100 mg $L^{-1}$ mixed standard solution. Working standard solutions (50, 25, 10, and 5 mg $L^{-1}$) were prepared by diluting the mixed standard solution with acetonitrile. To construct a calibration curve for quantification, each solution was diluted with acetonitrile to concentrations of 0.01, 0.02, 0.04, 0.1, 0.2, and 0.4 mg $L^{-1}$ and then diluted two-fold with acetonitrile or the untreated sample to prepare pure standards or matrix-matched standards of concentrations 0.005, 0.01, 0.02, 0.05, 0.1, and 0.2 mg $L^{-1}$, respectively.

## Sample preparation

For residue analysis of the test pesticide, the QuEChERS method was optimized. Two different extraction methods were used, viz. AOAC Official Method 2007.01, which uses acetate buffering (6 g MgSO$_4$, 1.5 g NaOAc), and the European Standard EN 15662 method, which uses citrate buffering (4 g MgSO$_4$, 1 g NaCl, 1 g Na$_3$Cit·2H$_2$O, 0.5 g Na$_2$HCit·1.5H$_2$O), and the recovery efficiency of each method was determined. Subsequently, the samples were purified using six different d-SPE tubes (Table 1), and the analytical efficiency of each type of d-SPE tube was determined.

To prepare the extracts, 10 g of each test crop (5 g for brown rice and soybean) was placed in a 50-mL conical centrifuge tube (Falcorn, USA). After adding 10 mL of acetonitrile to each

**Table 1. Compositions of the d-SPE tubes used for purification of extraction solutions.**

|        | MgSO$_4$ | PSA   | C$_{18}$ | GCB    |
|--------|----------|-------|----------|--------|
| d-SPE 1 | 150 mg   | 25 mg | -        | -      |
| d-SPE 2 | 150 mg   | 25 mg | 25 mg    | -      |
| d-SPE 3 | 150 mg   | 25 mg | -        | 2.5 mg |
| d-SPE 4 | 150 mg   | 50 mg | -        | -      |
| d-SPE 5 | 150 mg   | 50 mg | 50 mg    | -      |
| d-SPE 6 | 150 mg   | 50 mg | -        | 50 mg  |

PSA, *p*rimary secondary amine; GCB, graphitized carbon black.

tube, the samples were shaken for 5 min at 1,300 rpm. For the brown rice and soybean samples, extraction was performed after soaking the samples in 10 mL of distilled water for 1 h because hydrated rice and soybean samples provide better pesticide extraction results [13, 14]. After adding the two QuEChERS extraction pouches (one for acetate buffering and the other for citrate buffering), the extraction tube was shaken by hand for 30 s and then centrifuged at 3,500 rpm for 5 min to separate the aqueous and organic phases. Since soybeans contain a large amount of emulsifier, i.e., fat and protein [15], clear separation of the aqueous and organic layers was not achieved under these conditions and the amount of extracted solvent was small. Therefore, the soybean sample was centrifuged at 12,000 rpm for 10 min to improve the extraction efficiency. To purify the extracted samples, a 1-mL portion of the supernatant was added to each of the six d-SPE tubes (Table 1), which were then vortexed for ~30 s and centrifuged at 12,000 rpm for 5 min. The supernatant was diluted twice with acetonitrile for matrix matching and analyzed for test pesticides using the instrumental analysis method.

## Optimization of instrumental analysis

Since the test pesticide has twelve halogen atoms (one bromine atom and 11 fluorine atoms), analysis by gas chromatography with an electron capture detector (GC-ECD) is possible. However, to facilitate rapid and efficient pesticide residue analysis, LC-MS/MS was selected [8]. A reverse-phase octadecyl silica column (length = 150 mm, particle size = 2.7 μm) was used for effective separation of nonpolar and polar substances with a reasonable run time and peak resolution. The mobile phase was a 20:80 (v/v) mixture of distilled water and methanol with 0.1% formic acid as a protonation enhancer. To improve the selectivity and sensitivity for the test pesticides, sample analysis was conducted in the multiple reaction monitoring (MRM) mode. The MRM conditions were determined by performing a scan analysis of a standard solution (10 μg kg$^{-1}$) in infusion mode. The two most abundant ions were chosen as the quantitation and confirmation ions.

The precursor ion of broflanilide was observed at $m/z$ 665.0, and the quantitation and confirmation ions were observed at $m/z$ 556.0 and 506.1, respectively. The precursor ion of DM-8007 was observed at $m/z$ 648.9, and the quantitation and confirmation ions were observed at $m/z$ 242.1 and 77.2, respectively. The precursor ion of S(PFH-OH)-8007 was observed at $m/z$ 660.9, and the quantitation and confirmation ions were observed at $m/z$ 454.1 and 551.0, respectively. The optimized instrumental analysis conditions are shown in Table 2 and typical chromatograms for the analysis of broflanilide and its metabolites are shown in Fig 1.

## Method validation

The limit of quantitation (LOQ) of the analytical method was defined as the value obtained when the signal-to-noise ratio exceeded 10 and the reproducibility of the instrumental analysis was 5 μg kg$^{-1}$. To validate the pesticide residue analysis method, the reproducibility of the instrumental analysis and the recovery were investigated. The reproducibility of the instrumental analysis was verified by calculating the averages and relative standard deviations (RSDs) of the peak areas, peak heights, and retention times for standard solutions with concentrations of LOQ, 10LOQ, and 50LOQ. The recovery was evaluated by performing three repeated analyses of the untreated sample fortified with the standard solution at concentrations of 0.05 (10LOQ) and 0.25 (50LOQ) mg kg$^{-1}$.

## Matrix effect

The matrix effect (ME, %) is used to determine the effect of the analyte during ionization in the MS detector. The matrix-matched calibration method is a simple and effective method for

**Table 2. LC-MS/MS conditions for residual pesticide analysis in the test produce.**

| ⟨LC condition⟩ | |
|---|---|
| Instrument | Exion LC™, AB SCIEX, USA |
| Column | Halo $C_{18}$, 2.1 mm I.D. × 150 mm L. (2.7 μm particle size) |
| Flow rate | 0.2 mL min$^{-1}$ |
| Mobile phase | A: 5 mM ammonium formate in 0.1% formic acid (water based) |
| | B: 5 mM ammonium formate in 0.1% formic acid (MeOH based) |
| | A:B = 20:80 (v/v) |
| Injection volume | 1 μL |

| ⟨Mass condition⟩ | | | |
|---|---|---|---|
| Instrument | QTRAP 5500 system, AB SCIEX, USA | | |
| Ion spray voltage | 5500 V | Nebulizer gas | 50 psi |
| Curtain gas | 20 psi | Drying gas | 50 psi |
| Collision gas | 10 psi | Scan type | MRM mode |
| Drying temperature | 500°C | Ion source | ESI(+) |

| ⟨MRM condition⟩ | | | |
|---|---|---|---|
| pesticide | Q1 (*m/z*) | Q3 (*m/z*) | Collision energy (eV) |
| Broflanilide | 665.0 | 556.0 | 67 |
| | 665.0 | 506.1 | 81 |
| DM-8007 | 648.9 | 242.1 | 29 |
| | 648.9 | 77.2 | 129 |
| S(PFH-OH)-8007 | 660.9 | 454.1 | 77 |
| | 660.9 | 551.0 | 75 |

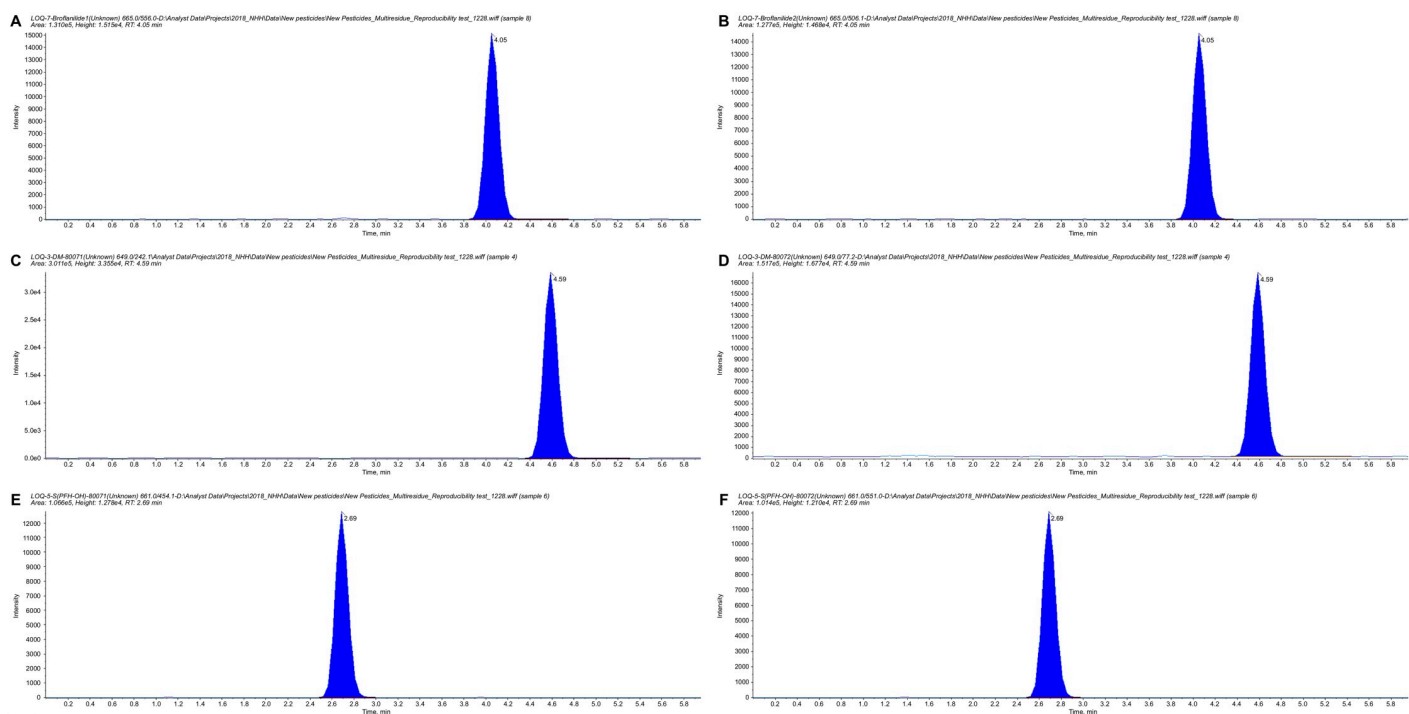

**Fig 1. Standard chromatograms for broflanilide and its two metabolites.** (A) Quantitation ion of broflanilide (665.0 → 556.0), (B) confirmation ion of broflanilide (665.0 → 506.1), (C) quantitation ion of DM-8007 (648.9 → 242.1), (D) confirmation ion of DM-8007 (648.9 → 77.2), (E) quantitation ion of S(PFH-OH)-8007 (660.9 → 454.1), and (F) confirmation ion of S(PFH-OH)-8007 (660.9 → 551.0).

offsetting the matrix effect [16–18]. To investigate the matrix effect in this study, calibration curves were constructed using the peak areas obtained by analyzing the pure standard and the matrix-matched standard. The matrix effect was calculated using the linear slopes, as shown in Eq (1) [19]. The matrix effect indicates the ion suppression or enhancement intensity, with $-20\% < ME < 20\%$, $-50\% < ME < -20\%$ or $20\% < ME < 50\%$, and $ME < -50\%$ or $ME > 50\%$ signifying low, medium, and high signal suppression or enhancement, respectively [19].

$$\text{ME (\%)} = \frac{\textit{Slope of matrix} - \textit{slope of matched standard calibration} - \textit{slope of pure standard calibration}}{\textit{Slope of pure standard calibration}} \times 100 \quad (1)$$

## Results and discussion

### Reproducibility test

The reproducibility of the instrument was confirmed by repeated analysis of the standard solutions at concentrations of LOQ, 10LOQ, and 50LOQ under the established instrumental analysis conditions (Table 3). The RSDs of the peak area, peak height, and retention time for broflanilide were less than 1.8%, 2.8%, and 0.6%, respectively. The corresponding values for DM-8007 were less than 2.4%, 3.6%, and 0.5%, and those for S(PFH-OH)-8007 were 3.1%, 3.5%, and 0.7%, respectively. These results indicated that the reproducibility of instrumental analysis was excellent [20].

Since excellent reproducibility was confirmed for the test pesticide, the test pesticide can be analyzed without using an internal standard. Furthermore, the analysis of the untreated samples confirmed the absence of any interfering substances that may hamper the analysis of the test pesticide and its metabolites.

### Matrix effect

In the test crop extracts obtained using the acetate and citrate buffering methods without purification, the matrix effects for broflanilide and its two metabolites ranged from −11.9% to 18.6% and from −13.5% to 12.1% (Fig 2). There were no significant differences between the extraction methods and signal suppression or enhancement was low. When matrix-matched standards were prepared by purifying the samples using six different d-SPE tubes after

**Table 3. Reproducibilities of the LC-MS/MS analysis of the test pesticides at concentrations of LOQ, 10LOQ, and 50LOQ.**

| Pesticide | Concentration (mg kg$^{-1}$) | Peak area | | Peak height | | Retention time (min) | |
|---|---|---|---|---|---|---|---|
| | | Mean ± SD | RSD (%) | Mean ± SD | RSD (%) | Mean ± SD | RSD (%) |
| Broflanilide | LOQ | 129,238 ± 2,282 | 1.8 | 14,829 ± 418 | 2.8 | 4.03 ± 0.0 | 0.4 |
| | 10LOQ | 1,304,875 ± 21,276 | 1.6 | 148,325 ± 3,275 | 2.2 | 4.04 ± 0.0 | 0.4 |
| | 50LOQ | 6,410,625 ± 110,164 | 1.7 | 721,413 ± 20,332 | 2.8 | 4.04 ± 0.0 | 0.6 |
| DM-8007 | LOQ | 287,463 ± 6,867 | 2.4 | 31,403 ± 1.121 | 3.6 | 4.60 ± 0.0 | 0.3 |
| | 10LOQ | 2,774,625 ± 25,528 | 0.9 | 301,088 ± 7,078 | 2.4 | 4.61 ± 0.0 | 0.4 |
| | 50LOQ | 13,666,250 ± 276,608 | 2.0 | 1,491,125 ± 49,133 | 3.3 | 4.60 ± 0.0 | 0.5 |
| S(PFH-OH)-8007 | LOQ | 109,375 ± 3,426 | 3.1 | 13,018 ± 394 | 3.0 | 2.70 ± 0.0 | 0.5 |
| | 10LOQ | 1,071,250 ± 19,091 | 1.8 | 125,625 ± 4,372 | 3.5 | 2.70 ± 0.0 | 0.4 |
| | 50LOQ | 5,335,000 ± 122,009 | 2.3 | 622,463 ± 16,129 | 2.6 | 2.70 ± 0.0 | 0.7 |

LOQ, limit of quantitation (5 μg kg$^{-1}$); SD, standard deviation; RSD, relative standard deviation.

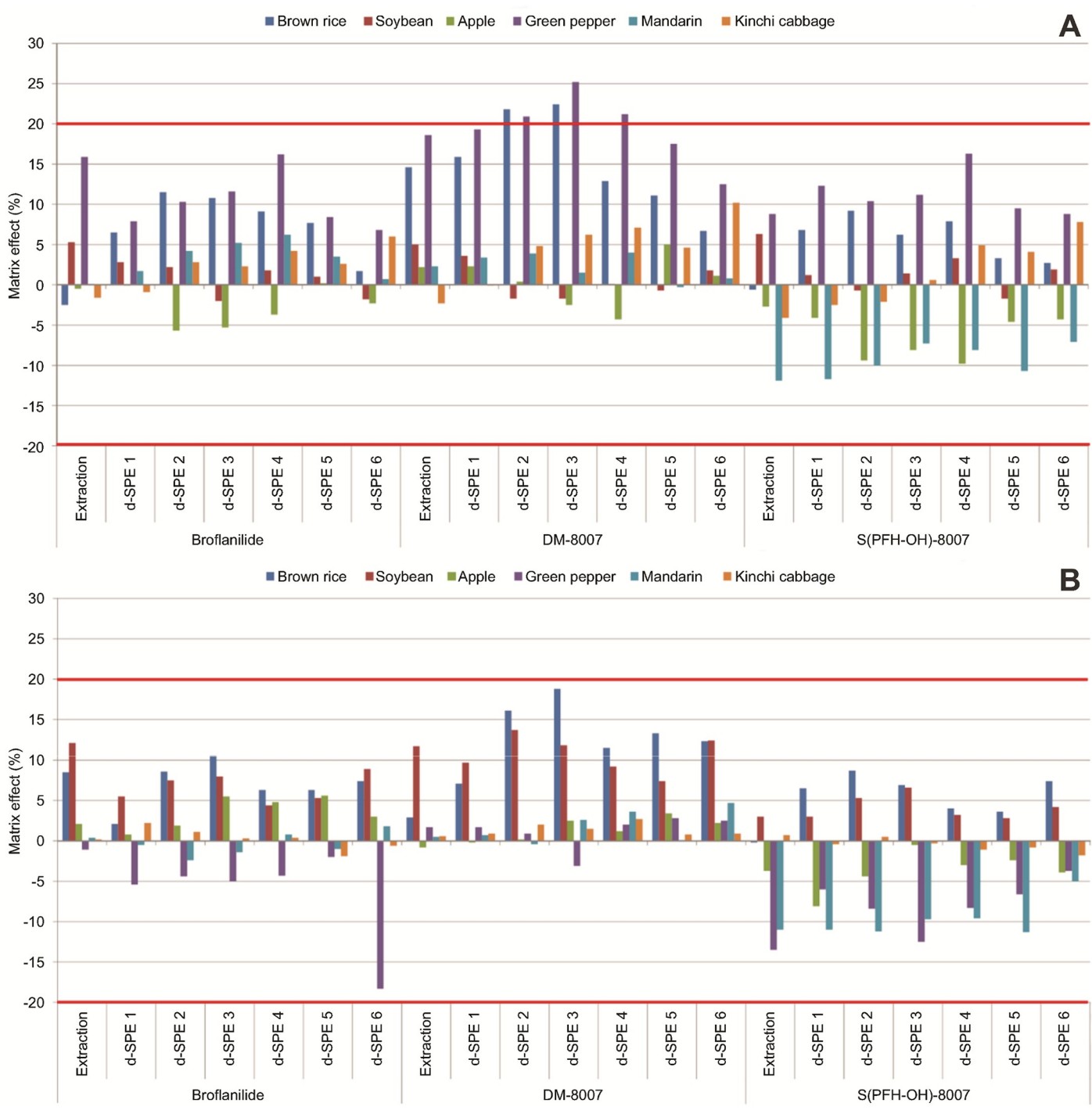

**Fig 2. Matrix effects of broflanilide and its two metabolites in the samples extracted using the (A) the acetate buffering method and (B) the citrate buffering method and purified by d-SPE using six different adsorbents.**

extraction following the acetate buffering method, the matrix effects were found to be low or medium (between −11.7% and 25.2%). In particular, those for DM-8007 in some brown rice and green pepper samples exceeded 20%. However, when the citrate buffering method was used, the matrix effects ranged from −18.3 to 18.8%, indicating low signal suppression or enhancement (Fig 2).

Various studies have been performed on reducing matrix effects [21–23], but Stahnke et al. [24] and Guo et al. [25] reported that the basic methods for removing impurities are dilution and purification. In this study, the purification process only reduced the matrix effects in a few samples. Since the matrix effect of the test pesticide in the test produce was so low, purification or dilution did not have a significant effect. This observation, which is similar to the results reported by Dušek et al. [26], indicated that the type or amount of adsorbent used had little influence on the matrix effect.

Although the matrix effects were found to be low, matrix-matched standards should be used to quantify test pesticides. Therefore, the recoveries of broflanilide and its metabolites in the test produce were calculated using matrix-matched standard calibration.

## Recovery test

The average recoveries of broflanilide, DM-8007, and S(PFH-OH)-8007 in the extracts obtained by acetate buffering without purification ranged from 97.2% ± 1.7% to 111.7% ± 2.5%, from 96.2% ± 2.0% to 108.7% ± 1.4%, and from 96.7% ± 2.0% to 105.6% ± 1.4%, respectively. The average recoveries of broflanilide, DM-8007, and S(PFH-OH)-8007 in the extracts obtained using citrate buffering without purification ranged from 97.8% ± 2.2% to 102.9% ± 0.4%, from 83.0% ± 3.7% to 106.6% ± 1.2%, and from 95.9% ± 0.9% to 105.7% ± 0.9%, respectively (Fig 3). Since all the recoveries were within the valid recovery range of 70–120%, both extraction methods can be considered suitable for the extraction of residual pesticides in the test produce materials [27].

The average recoveries of broflanilide extracted by either method and purified by d-SPE using 150 mg $MgSO_4$, 25 mg PSA, and $C_{18}$ or GCB ranged from 99.4% ± 2.9% to 113.7% ± 3.5% and from 98.6% ± 0.3% to 114.1% ± 2.9%, respectively. The average recoveries of DM-8007 and S(PFH-OH)-8007 using the same methods ranged from 92.6% ± 1.9% to 110.3% ± 2.5% and from 74.1% ± 0.5% to 116.9% ± 0.9%, respectively (Fig 3). Here, the recoveries did not depend significantly on the type of adsorbents used (e.g., $C_{18}$ or GCB). The average recoveries of broflanilide extracted by either method and purified by d-SPE using 150 mg $MgSO_4$, 50 mg PSA, and $C_{18}$ or GCB ranged from 98.6% ± 1.7% to 108.6% ± 0.5% and from 94.4% ± 1.0% to 111.0% ± 1.0%, respectively. The average recovery of S(PFH-OH)-8007 fortified at 0.05 mg $kg^{-1}$ in brown rice extracted by either method and purified using d-SPE 6 (150 mg $MgSO_4$, 50 mg PSA, and 50 mg GCB) was 69.4% ± 1.6%. This value was somewhat low, but all the other purification methods met the required recovery range of 70–120%. However, for apples, the average recoveries of DM-8007 extracted by the acetate or citrate buffering methods and purified using d-SPE 6 mixed (150 mg $MgSO_4$, 50 mg PSA, and 50 mg GCB) ranged from 53.9% ± 0.5% to 55.8% ± 0.7% and from 59.3% ± 1.4% to 63.6% ± 4.8%, respectively. Similarly, the corresponding values for kimchi cabbage ranged from 59.3% ± 0.3% to 61.0% ± 0.9% and from 54.0% ± 1.9% to 54.5% ± 1.5%, which were outside the effective recovery range of 70–120%. These results were the same as those reported by Li et al., who found that the samples purified using a large amount of GCB may have a low recovery [28].

An et al. [8], who examined various types and amounts of adsorbents, reported that the recoveries of broflanilide and its two metabolites were the best when a combination of 50 mg PSA and 10 mg GCB was used for purification. They also reported that the amount of GCB had the greatest effect on the recovery. Guo et al. [25] reported that the recovery of some pesticides decreased when more than 10 mg of GCB was used for purification. Other studies have shown that the recovery decreases with increasing amounts of GCB, which is used to remove pigment and sterol compounds [7]. In particular, GCB is effective in removing pigments [4] from analytes (Fig 4). Therefore, the amount of GCB applied should be varied depending on

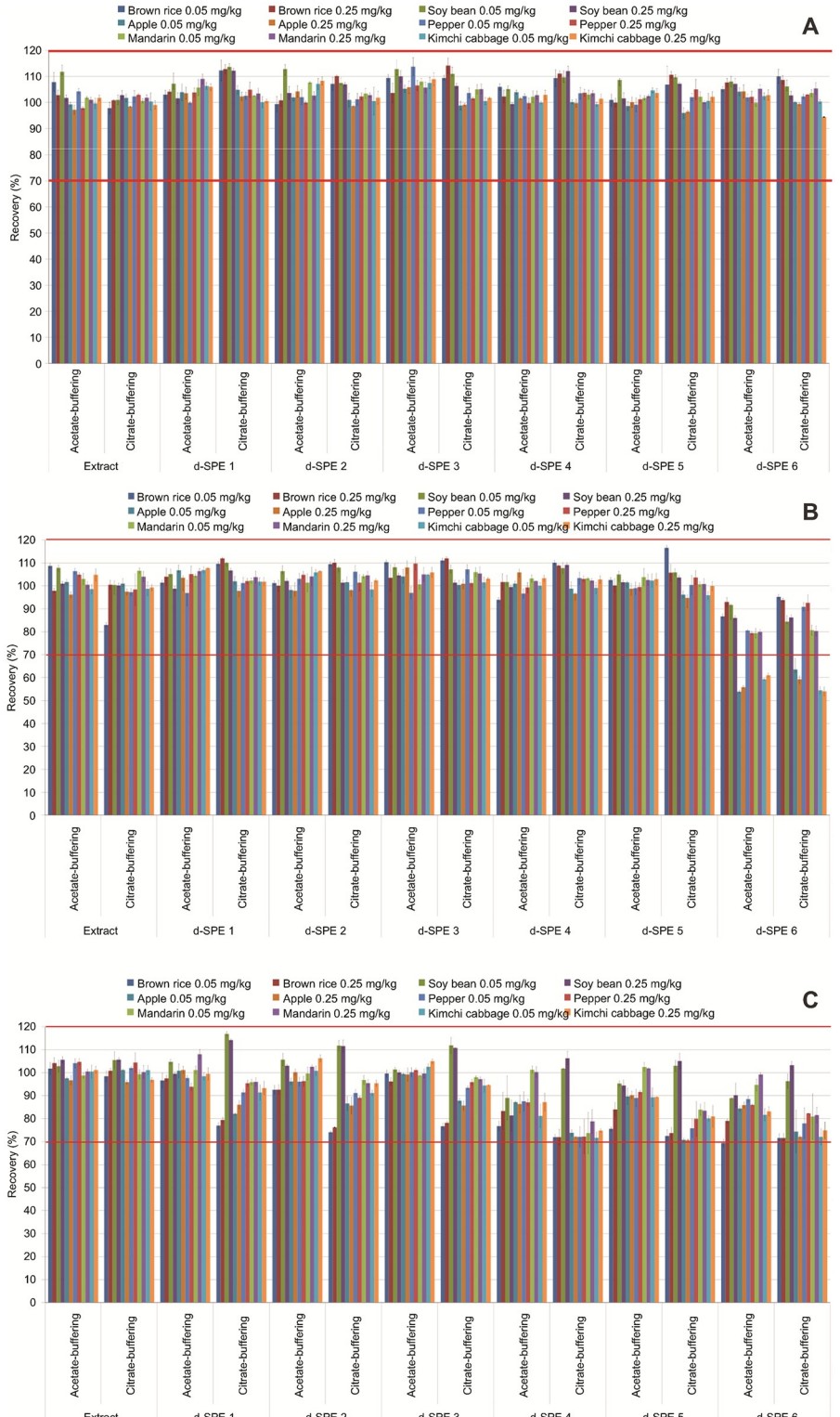

**Fig 3. Recoveries of (A) broflanilide, (B) DM-8007, and (C) S(PFH-OH)-8007 from six test crops extracted by the acetate and citrate buffering methods and purified by d-SPE using six different adsorbents.**

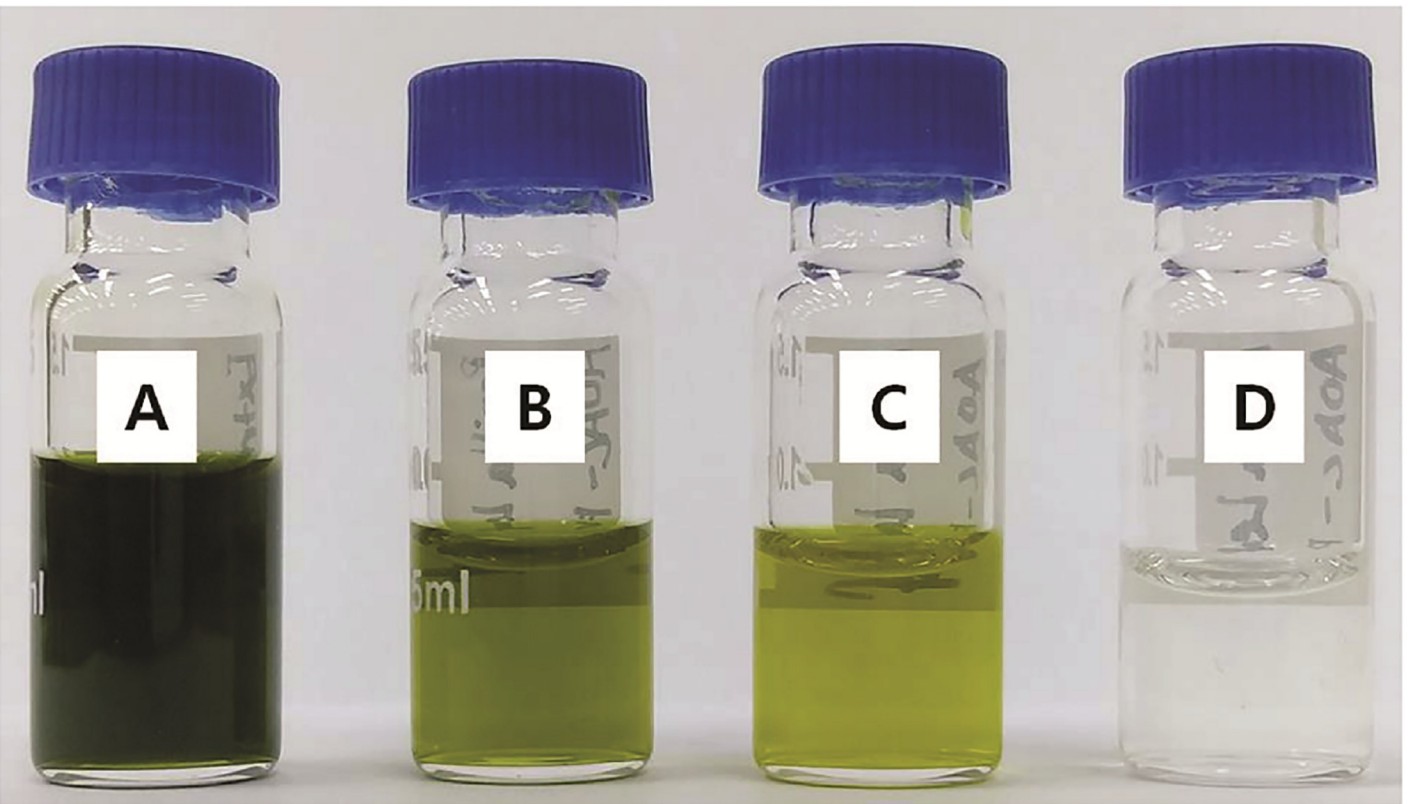

**Fig 4. Sample colors of a pesticide residue extract purified by d-SPE using different adsorbent compositions.** (A) Sample obtained by extraction using the acetate buffering method, (B) extracted sample after purification by d-SPE using 150 mg MgSO$_4$ and 50 mg PSA, (C) extracted sample after purification by d-SPE using 150 mg MgSO$_4$, 50 mg PSA, and 50 mg C$_{18}$, and (D) extracted sample after purification by d-SPE using 150 mg MgSO$_4$, 50 mg PSA, and 50 mg GCB.

the matrix. Finally, considering the recovery and matrix effect results obtained herein, a reliable residue analysis for broflanilide and its two metabolites was achieved when extracts were obtained using the citrate buffering method and were purified by d-SPE using 25 mg PSA with other adsorbents.

## Conclusions

In this study, a pesticide residue analysis method was developed for the detection and quantification of broflanilide and its two metabolites, DM-8007 and S(PFH-OH)-8007, in various crops. In particular, the effects of the QuEChERS extraction method and purification adsorbent were evaluated. Purification using the adsorbents PSA, C18, and GCB did not significantly reduce the matrix effect because the matrix effect was already low before purification, as reported by Dušek et al. [26]. The lowest matrix effects were observed for samples extracted by the citrate buffering method and purified by d-SPE using 25 mg PSA. Thus, this method is suitable for residue analysis of broflanilide and its two metabolites. The recovery of broflanilide was excellent, regardless of the adsorbent type and amount. However, although the recovery of S(PFH-OH)-8007 was adequate under all the tested purification conditions, it decreased with increasing adsorbent amount. Moreover, the recoveries of DM-8007 were low in apple and kimchi cabbage purified using 50 mg of GCB. Thus, the major factor affecting the recovery of metabolite DM-8007 was the type and amount of adsorbents used in the purification process [25]. In particular, although GCB is effective in removing pigments [4] from analytes, its

application should be varied depending on the crop because it may reduce the recovery [29]. In a study on the determination of broflanilide and its two metabolites in five soils, An et al. [8] found that GCB mixed with another adsorbent was more effective for purification than only $C_{18}$ or PSA. Therefore, broflanilide and its two metabolites can be determined as residual pesticides in various crops by modifying the adsorbent composition for purification by d-SPE in the QuEChERS method.

## Author Contributions

**Conceptualization:** Danbi Kim.

**Data curation:** Hyeyoung Kwon, Kee Sung Kyung.

**Formal analysis:** Hyun Ho Noh.

**Funding acquisition:** Danbi Kim.

**Investigation:** Hyun Ho Noh, Chang Jo Kim, Byeong-chul Moon, Sujin Baek, Min-seok Oh.

**Methodology:** Danbi Kim.

**Project administration:** Danbi Kim.

**Resources:** Danbi Kim.

**Software:** Hyun Ho Noh.

**Supervision:** Danbi Kim.

**Validation:** Hyun Ho Noh, Chang Jo Kim, Byeong-chul Moon, Sujin Baek, Min-seok Oh.

**Visualization:** Hyun Ho Noh.

**Writing – original draft:** Hyun Ho Noh.

**Writing – review & editing:** Hyun Ho Noh, Danbi Kim.

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
