## [Decision Letter · Decision Letter 0]

14 Jul 2020

PONE-D-20-18490

Optimized residue analysis method for broflanilide and its metabolites in agricultural produce using the QuEChERS method and LC-MS/MS

PLOS ONE

Dear Dr. Kim,

Thank you for submitting your manuscript to PLOS ONE. After careful consideration, we feel that it has merit but does not fully meet PLOS ONE’s publication criteria as it currently stands. Therefore, we invite you to submit a revised version of the manuscript that addresses the points raised during the review process.

We look forward to receiving your revised manuscript.

Kind regards,

Ch Ratnasekhar, Ph.D.

Academic Editor

PLOS ONE

Journal Requirements:

2. In your Methods, please state the exact source of the samples used in your study

Reviewers' comments:

Reviewer's Responses to Questions

**Comments to the Author**

1. Is the manuscript technically sound, and do the data support the conclusions?

Reviewer #1: Yes

Reviewer #2: Partly

2. Has the statistical analysis been performed appropriately and rigorously? 

Reviewer #1: N/A

Reviewer #2: Yes

3. Have the authors made all data underlying the findings in their manuscript fully available?

Reviewer #1: Yes

Reviewer #2: Yes

4. Is the manuscript presented in an intelligible fashion and written in standard English?

Reviewer #1: Yes

Reviewer #2: Yes

5. Review Comments to the Author

Reviewer #1: You have two main things to fix. First, is the comment and appropriate response for failing to include an internal standard. Second, the lack of matrix effects should likely not be reported, since there was little matrix effect. Maybe you can add that in in some minor way, but not the way it has been written.

Reviewer #2: The manuscript deals with a subject of interest and is technically correct from an analytical point of view. However, authors did not apply the developed method for the quantitative analysis in real samples. Thus it is better to include that part in this MS to show the applicability.

6. PLOS authors have the option to publish the peer review history of their article (what does this mean?). If published, this will include your full peer review and any attached files.

Reviewer #1: **Yes: **Brian Quinn

Reviewer #2: No

---

## [Author Response · Author response to Decision Letter 0]

26 Aug 2020

Response to Reviewers

Reviewer 1.

Comment 1. Why didn’t you include the internal standard?

Answer: Internal standards are used to reduce the matrix effect. However, the matrix effect observed in our study was insignificant, i.e., did not have to be reduced via internal standard addition. In the future, with the above comment in mind, we will investigate the effects of internal standard addition.

Comment 2. There is little matrix effect, so you don’t have to report it.

Answer: In most cases, the matrix effect was low, but in some cases it was moderate. Therefore, we provided results for various cases to suggest that an appropriate method can be selected for the analysis of broflanilide and its metabolites in agricultural products.

Reviewer 2.

Comment 1. It is recommended to present the analysis results by applying the established analysis method to the real samples.

Answer: We totally agree with this comment. Agricultural products in circulation in Korea are monitored by certain institutions for pesticide residues. In particular, our agency aims to develop analytical methods (such as that described in the present work) that can be used by these institutions. So, we not carried out to monitor pesticide residue in circulating agricultural products.

Editor comments

Comment 1. Please ensure that your manuscript meets PLOS ONE's style requirements, including those for file naming.

Answer: The necessary changes have been made: the excessive use of acronyms has been avoided, references have been formatted, etc.

Comment 2. In your Methods, please state the exact source of the samples used in your study.

Answer: The untreated samples were purchased from an environmentally friendly agricultural produce market, Chorocmaeul (www.choroc.com) in Wanju, Korea. (Page5, Line100)

Comment 3. PLOS requires an ORCID iD for the corresponding author in Editorial Manager on papers submitted after December 6th, 2016. Please ensure that you have an ORCID iD and that it is validated in Editorial Manager.

Answer: An ORCID iD for the corresponding author has been successfully created and validated.

---

## [Decision Letter · Decision Letter 1]

22 Sep 2020

Optimized residue analysis method for broflanilide and its metabolites in agricultural produce using the QuEChERS method and LC-MS/MS

PONE-D-20-18490R1

Dear Dr. Danbi Kim,

We’re pleased to inform you that your manuscript has been judged scientifically suitable for publication and will be formally accepted for publication once it meets all outstanding technical requirements.

Kind regards,

Ch Ratnasekhar, Ph.D.

Academic Editor

PLOS ONE

Additional Editor Comments (optional):

Reviewers' comments:

Reviewer's Responses to Questions

**Comments to the Author**

1. If the authors have adequately addressed your comments raised in a previous round of review and you feel that this manuscript is now acceptable for publication, you may indicate that here to bypass the “Comments to the Author” section, enter your conflict of interest statement in the “Confidential to Editor” section, and submit your "Accept" recommendation.

Reviewer #2: All comments have been addressed

2. Is the manuscript technically sound, and do the data support the conclusions?

Reviewer #2: Yes

3. Has the statistical analysis been performed appropriately and rigorously? 

Reviewer #2: N/A

4. Have the authors made all data underlying the findings in their manuscript fully available?

Reviewer #2: Yes

5. Is the manuscript presented in an intelligible fashion and written in standard English?

Reviewer #2: Yes

6. Review Comments to the Author

Reviewer #2: Authors have satisfactorily answered all queries raised by reviewers. Now this manuscript may be accepted to publish in this journal.

7. PLOS authors have the option to publish the peer review history of their article (what does this mean?). If published, this will include your full peer review and any attached files.

Reviewer #2: No

---

## [Editor Report · Acceptance letter]

28 Sep 2020

PONE-D-20-18490R1 

Optimized residue analysis method for broflanilide and its metabolites in agricultural produce using the QuEChERS method and LC-MS/MS 

Dear Dr. Kim:

I'm pleased to inform you that your manuscript has been deemed suitable for publication in PLOS ONE. Congratulations! Your manuscript is now with our production department. 

Kind regards, 

on behalf of

Dr. Ch Ratnasekhar 

Academic Editor

PLOS ONE